# How Much is Enough?
# The Diminishing Returns of Tokenization Training Data

**Varshini Reddy** [1]   **Craig W. Schmidt** [1]   **Yuval Pinter** [2]   **Chris Tanner** [1][3]

## Abstract

Tokenization, a crucial initial step in natural language processing, is governed by several key parameters, such as the tokenization algorithm, vocabulary size, pre-tokenization strategy, inference strategy, and training data corpus. This paper investigates the impact of an often-overlooked hyperparameter, tokenizer training data size. We train BPE, UnigramLM, and WordPiece tokenizers across various vocabulary sizes using English training data ranging from 1GB to 900GB. Our findings reveal diminishing returns as training data size increases beyond roughly 150GB, suggesting a practical limit to the improvements in tokenization quality achievable through additional data. We analyze this phenomenon and attribute the saturation effect to constraints introduced by the pre-tokenization stage. We then demonstrate the extent to which these findings can generalize by experimenting on data in Russian, a language typologically distant from English. For Russian text, we observe diminishing returns after training a tokenizer from 200GB of data, which is approximately 33% more than when training on English. These results provide valuable insights for optimizing the tokenization process by reducing the compute required for training on large corpora and suggest promising directions for future research in tokenization algorithms.

## 1. Introduction

Tokenizers are a foundational component of any NLP pipeline, as they are responsible for converting raw text into useful sequences of indexed tokens. Practitioners often default to standard tokenization algorithms such as Byte-Pair Encoding (BPE; Sennrich et al., 2016), UnigramLM (Kudo, 2018) or WordPiece (Schuster & Nakajima, 2012; Devlin et al., 2019), sourced directly from libraries such as Hugging Face.[1] The training process of a tokenizer involves using a corpus of training data and a specific tokenization algorithm to generate a fixed-size vocabulary, usually containing between 32,000 and 128,000 tokens in the monolingual case.

While extensive research explores the influence of a *model's* training data on LLM performance (Pearce et al., 2024; Zhang et al., 2024; Hoffmann et al., 2022; Kaplan et al., 2020), the impact of a *tokenizer's* training data remains relatively unexplored. Recent work has begun to address the importance of tokenizers' vocabulary sizes and the training data domains (Dagan et al., 2024), tying into existing exploration of other aspects of tokenization, including the influence of different tokenization algorithms (Schmidt et al., 2024; Ali et al., 2024; Mielke et al., 2021; Rai & Borah, 2021), vocabulary size optimization (Gowda & May, 2020), and the interplay between data type and tokenization strategy, especially in multi-lingual applications (Limisiewicz et al., 2023; Rust et al., 2021). Yet, to the best of our knowledge, we are the first to investigate how much training data is needed for a tokenizer, and how this affects performance.

We address this by examining the impact of scaling tokenizer training data with sizes ranging from 1GB to 900GB. We train English BPE, UnigramLM, and WordPiece tokenizers with vocabulary sizes of 40,960, 64,000, 128,000, and 256,000. For each variant, we examine the proportion of vocabulary tokens that are shared with the 900GB reference case, and use intrinsic metrics to measure the quality of tokenization on a held-out evaluation corpus of 150GB. Our findings indicate that increasing the amount of tokenizer training data leads to diminishing returns in the compute-versus-gain tradeoff, suggesting a saturation point at roughly 150GB of training data beyond which further data provides minimal to no improvements in tokenization quality. We then examine the proportion of pre-tokenization chunks that exactly match a single token in the tokenizer vocabulary, and suggest that this very high proportion is a possible explanation for these diminishing returns.

[1]Kensho Technologies, Cambridge, MA [2]Ben-Gurion University of the Negev, Beer Sheva, Israel [3]Massachusetts Institute of Technology, Cambridge, MA. Correspondence to: Varshini Reddy <varshini.bogolu@kensho.com>.

*Proceedings of the ICML 2025 Tokenization Workshop (TokShop)*, Vancouver, Canada. PMLR 267, 2025. Copyright 2025 by the author(s).

[1]https://github.com/huggingface/tokenizers

Following up on the English experiments, we turn to Russian as a second test case to see whether the main findings hold up on a language that is distinct from English both by using a different alphabet and by various typological properties such as freedom in word-order and morphological richness. We focus on BPE vocabularies of size 40,960 ranging from 30GB to 600GB, identifying a later saturation point than in English at around 200GB, suggesting that complexity in word formation may impact the data hunger of tokenizers, requiring them to take up more samples from a larger and more idiosyncratic word-level vocabulary.

## 2. Effect of Training Corpus Size on Tokenization

Our training corpus for the main experiments in English combines the de-duplicated PILE (Gao et al., 2020) and RedPajama (Weber et al., 2024) datasets, totaling 900GB of text.[2] In addition, we held out a de-duplicated set of size 150GB from the PILE and RedPajama corpus for evaluation.

We build token vocabularies using three algorithms: BPE,[3] UnigramLM,[4] and WordPiece.[5] For each algorithm, we train vocabularies of sizes 40K, 64K, 128K, and 256K, on progressively larger subsets of the randomly shuffled 900GB corpus. In order to evaluate the effects of training data size on both smaller and larger scales, we start with finer increments of 1GB, 5GB, and 15GB subsets, followed by all 30GB increments between 30GB and the full 900GB corpus. The size increase is augmentative, meaning that each corpus subset properly includes all those with smaller size. Overall, we trained 33 distinct vocabularies for each of the four vocabulary sizes for each of the three tokenizer algorithms, leading to a total of 396 trained vocabularies for English.

**Pre-processing** To facilitate training token vocabularies on datasets ranging to hundreds of gigabytes, we implemented a parallel pre-tokenization step, which used a regular expression to break the documents into chunks. We then aggregated the counts of each pre-tokenized chunk over all documents. This pre-tokenized corpus of chunks and counts served as the input for all subsequent vocabulary building, significantly reducing its computational overhead by avoid-

ing tokenization of common chunks more than once. All tokenizer code was modified to support this method of aggregate chunking and incremental counting. Listing 1 shows the regular expression used by GPT-4 for pre-tokenization, which we also used for our parallel pre-tokenizer. A breakdown of this regex, with an explanation of each branch, is presented in Appendix A.

```
r"(?i:[sdmt]|ll|ve|re)|[^\r\n\p{L}\p{N}]?+\
    p{L}+|\p{N}{1,3}|_?[^\s\p{L}\p{N}]++[\r
    \n]|\s[\r\n]|\s+(?!\S)|\s+"
```

Listing 1: GPT-4 pre-tokenizer regular expression.

### 2.1. Vocabulary analysis

Figure 1 illustrates the fraction of vocabulary shared between tokenizers trained on varying amounts of data and a reference of the same algorithm trained on the full 900GB corpus, for vocabulary sizes ranging from 40,960 to 256,000. This fraction increases consistently with the amount of training data. For a vocabulary size of 40,960, the shared vocabulary rises from approximately 58% to 97% for BPE, from 40% to 97% for UnigramLM, and from 4% to 92% for WordPiece. This trend of increased shared vocabulary with larger training datasets is observed across all tested vocabulary sizes. A more detailed comparison is provided in Appendix B.

As vocabulary size increases from 40,960 to 256,000, the proportion of common vocabulary stabilizes at a higher value for larger datasets. BPE and UnigramLM tokenizers exhibit similar convergence patterns, with BPE consistently maintaining a slightly higher proportion of common vocabulary across all vocabulary sizes. WordPiece demonstrates a more gradual increase in shared vocabulary, particularly at larger vocabulary sizes, indicating a greater sensitivity to increasing data volume. Although initial fluctuations are more pronounced at a vocabulary size of 256,000, all tokenizers eventually converge as data scales. These results collectively suggest that while larger training datasets consistently yield more similar vocabularies across algorithms, BPE and UnigramLM exhibit more rapid convergence than WordPiece, perhaps due to the sensitivity afforded in its merge selection criterion to token pairs which are relatively rare individually but co-occur frequently. Such pairs are likely to shift erratically as data is added, with more co-occurrences of such pairs surfacing, increasing their pairwise fit score at a rate disproportional to that of the overall data.

Finally, we raise the practical implication from our findings, that a substantial portion of the vocabulary learned from the full 900GB dataset can be effectively obtained from tokenizers trained on significantly smaller fractions of the data. At this point, we cannot ascertain whether the remaining differences in vocabulary are artifacts of small-

---

[2]As our work is the first of its kind in the tokenization space, there was no predefined baseline for corpus size to reference. However, a recent study (Schmidt et al., 2024) used 200 billion tokens (approximately 850GB of the PILE dataset) for tokenizer comparisons, which we adopted as our upper limit. This choice serves as a practical reference point rather than an assumed optimal baseline.

[3]We use a custom BPE tokenizer based on `https://github.com/karpathy/minbpe` as a starting point.

[4]`https://github.com/huggingface/tokenizers/blob/main/bindings/python/py_src/tokenizers/implementations/sentencepiece_unigram.py`

[5]`https://github.com/huggingface/tokenizers/blob/main/bindings/python/py_src/tokenizers/implementations/bert_wordpiece.py`

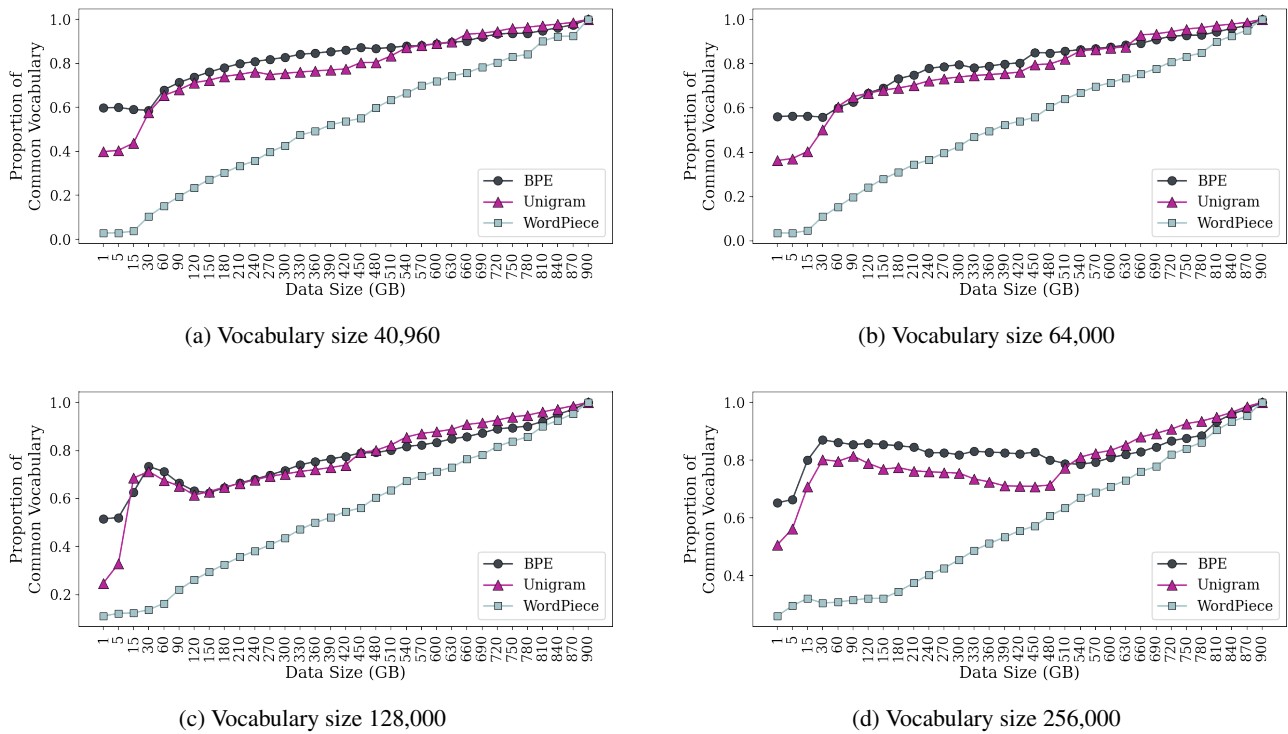

(a) Vocabulary size 40,960

(b) Vocabulary size 64,000

(c) Vocabulary size 128,000

(d) Vocabulary size 256,000

Figure 1: Proportion of common vocabulary for BPE, UnigramLM, and WordPiece tokenizers, trained with cumulatively increasing data, relative to the vocabulary of the corresponding tokenizer trained with 900GB of data.

vocabulary underfitting or large-vocabulary overfitting. It stands to reason that, if the vocabulary is meant to be used in variable or unknown domains, smaller might be better, whereas a *trusted* large corpus might benefit downstream applications which are more well-defined. We thus turn to comparative evaluation of token vocabulary behavior in different settings and across different corpora. In §4, we further expand this inquiry of generalization by repeating key parts of the analysis on data in Russian.

### 2.2. Intrinsic analysis

While we have seen that there are significant differences in the vocabularies of tokenizers trained on, for example, 30GB versus 900GB of data, it is important to examine these differences to determine whether they translate into meaningful improvements in tokenization quality. To evaluate the trained tokenizers without the additional computational overhead of training dozens of full LLMs, we use the intrinsic tokenizer metrics collected by Uzan et al. (2024). Following prior work (Schmidt et al., 2024; Zouhar et al., 2023), we interpret these independent intrinsic measures summarized below as a proxy for the hypothetical performance of a tokenizer on downstream tasks.

**Morphological alignment:** Measures how well a tokenizer's word segmentations match gold-standard morpho-logical segmentations. Higher scores indicate a greater ability to capture word structure.

**Cognitive score:** Assesses the correlation between a tokenizer's output and human performance in lexical decision tasks, evaluating how well the tokenizer's behavior aligns with human lexical processing (Beinborn & Pinter, 2023). Specifically, this evaluation dataset examines the correlation between the degree to which tokenizers segment character sequences and the ability of humans to recognize them as words.

**Rényi efficiency:** Encourages optimal encoding by evaluating the ratio of the Shannon entropy to the maximum possible entropy of the token distribution. Higher entropy indicates a more diverse and unpredictable distribution, penalizing vocabularies which are skewed towards very frequent and/or very rare tokens. We use $\alpha = 2.5$ for our analysis following Zouhar et al. (2023).

Figure 2 presents the intrinsic metric results for BPE across the four vocabulary sizes included in our study, evaluated on a 150 GB held-out dataset sampled from our corpus. Contrary to our expectations, these intrinsic measures do not reveal substantial performance gains with increasing data size. In fact, for BPE, performance on all metrics plateaus in the 120GB to 150GB range. Despite the observed vocab-

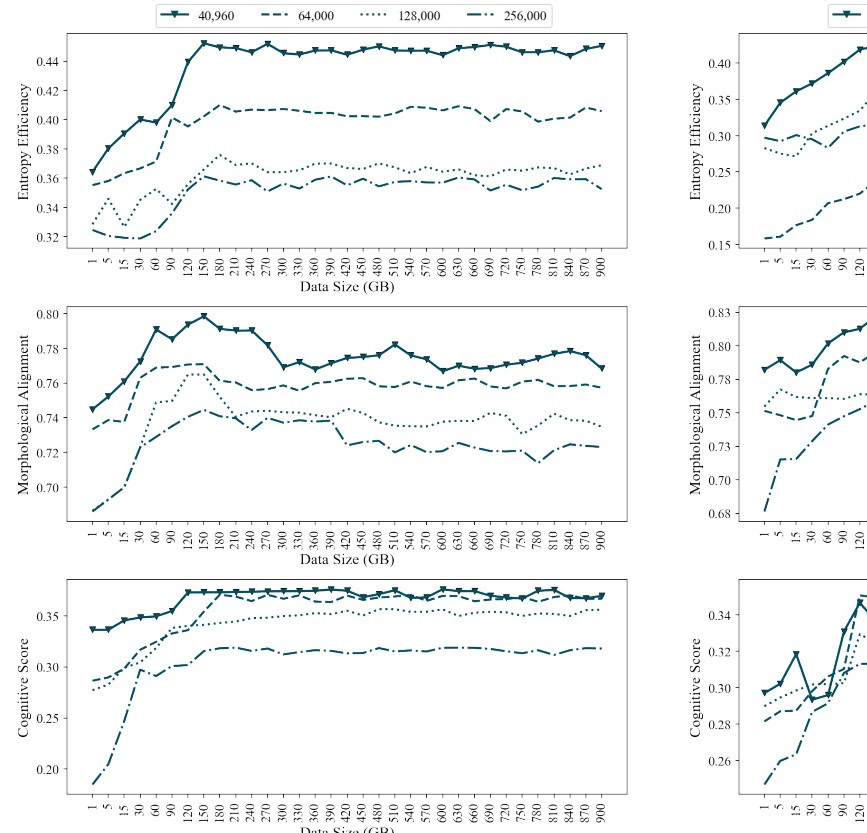 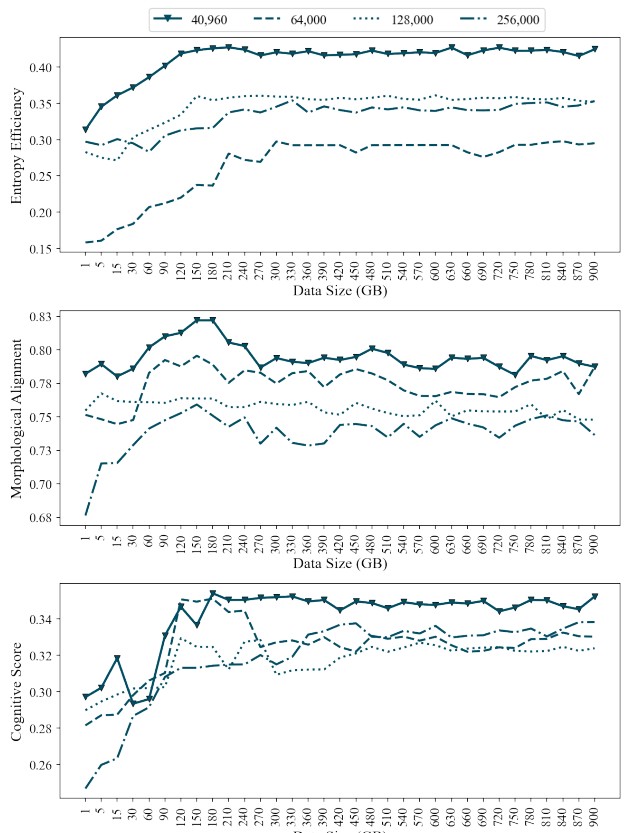

Figure 2: Intrinsic measures on our held-out set for **BPE** tokenizers trained for each of the four vocabulary sizes with varied training corpus size.

Figure 3: Intrinsic measures of **UnigramLM** tokenizers trained for each of the four vocabulary sizes with scaled training data.

ulary shifts, the core properties reflected by these metrics remain relatively stable across the range of training data volumes. Thus, simply increasing the training data size may not inherently lead to substantial improvements in a tokenizer's effectiveness.

Figure 3 displays benchmark results for the UnigramLM tokenizers trained with varying data sizes and vocabulary sizes. Similar to the observations with BPE, the UnigramLM tokenizers do not show a substantial and consistent improvement in intrinsic scores with increasing data size. While the cognitive score score exhibits a slight upward trend initially, it plateaus beyond approximately 180GB, suggesting that further increases in training data do not significantly enhance the tokenizer's alignment with morphological segmentations. The average F1 score calculated over the morphological benchmarks, while showing some fluctuations, does not demonstrate a clear and sustained improvement with larger datasets. The entropy score, similar to the other metrics, plateaus relatively early, indicating that the balance of token frequencies stabilizes with increasing data size.

For WordPiece (Figure 4), a similar pattern emerges. While

slightly higher variations exist, particularly in cognitive score, the intrinsic scores do not substantially improve with increasing data size.

### 2.3. Cross-domain analysis

To reconcile the apparent discrepancy between observed vocabulary shifts and stable intrinsic scores, we analyzed the impact of our trained tokenizers on a diverse, multi-domain evaluation dataset. This data spans several domains: *biology, code, finance, history, legal, mathematics, general text*. This diverse composition mitigates potential tokenizer biases that arise from the influence of domain-specific vocabulary prevalence. Each domain-specific corpus contains 1.5 million characters. While the sources of the evaluation sets are presented in Appendix C, Figure 5 depicts the composition of terms across specific domains. Each data point represents a proportion of general English terms, numerical values, and domain-specific terms.

We assessed the impact of scaling training data for vocabulary construction on the evaluation set by computing the Jaccard Index between the actual tokens used in the eval-

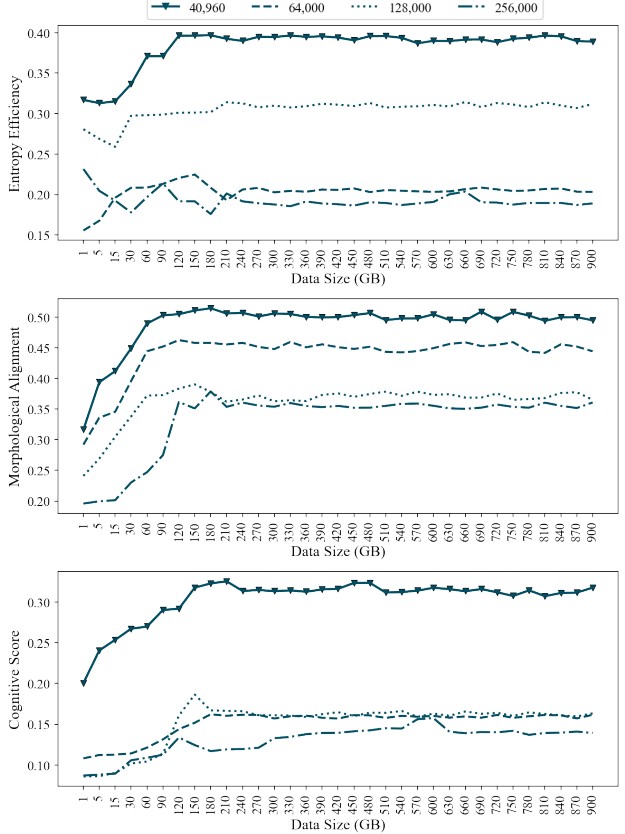

Figure 4: Intrinsic measures of **WordPiece** tokenizers trained for each of the four vocabulary sizes with scaled training data

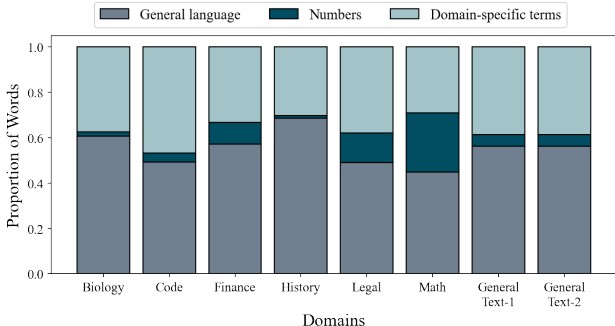

Figure 5: Proportion of general English terms, domain-specific terms, and numerical values within each evaluation set.

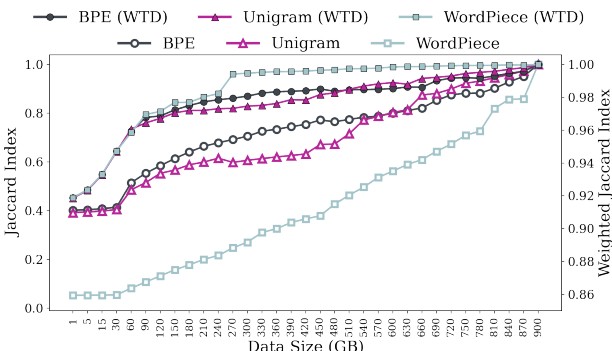

Figure 6: Jaccard Index (open markers) and Weighted Jaccard Index (WTD; filled markers) for BPE, UnigramLM, and WordPiece tokenizers (vocabulary size **40,960**) across varying data sizes, averaged over all evaluation domains.

uation text using each trained tokenizer and the same text tokenized with the 900GB-trained reference tokenizer:

$$J(U, V) = \frac{|U \cap V|}{|U \cup V|}, \qquad (1)$$

where $U$ and $V$ are the vocabularies of the reference tokenizer and the current tokenizer.

We also computed a weighted version of the Jaccard Index using the normalized token counts over the evaluation data, to account for the differences in the training set sizes:

$$J_w(U, V) = \frac{\sum_{t \in U \cap V} \min(w_U(t), w_V(t))}{\sum_{t \in U \cup V} \max(w_U(t), w_V(t))}, \qquad (2)$$

where $w_U(t)$ and $w_V(t)$ are the normalized token frequencies for token $t$ in vocabularies $U$ and $V$.

Figure 6 presents the standard (open markers) and weighted (filled markers) Jaccard Index for token vocabularies of size 40,960. The weighted scores consistently exceed the unweighted scores, highlighting the significant influence of token frequency on vocabulary overlap. This suggests that

high-frequency tokens exhibit greater consistency across varying data sizes, and they account for a significant fraction of the training token counts. Again the unique behavior of WordPiece surfaces, demonstrating its preference for tokens which co-occur strongly but at an overall low frequency, leading to large observable differences in vocabulary sets that barely translate to effects at the corpus level.

Our analysis revealed that over 80% of our evaluation text is represented by approximately 20% of the tokenizer vocabulary (see Appendix C for detailed results across individual domains of our evaluation dataset and vocabulary sizes) This finding suggests that the majority of tokens that are added with increasing training data are low-frequency and thus less consequential. While the specific composition of the vocabulary evolves with an increase in training data, the core set of tokens responsible for representing the majority of the text remains relatively stable. This observation is consistent with Zipf's law (Zipf, 1949), which postulates an inverse relationship between word frequency and rank in natural language corpora. Thus, adding to a tokenizer's training

data beyond a certain point primarily adds low-frequency tokens to the vocabulary, which has a limited impact on the overall tokenization characteristics captured by the intrinsic metrics. This is one explanation as to why the intrinsic metrics are largely unaffected by increases in training data.

## 3. The Limiting Role of Pre-Tokenization

Pre-tokenization is the initial step in the tokenization process, where regular expressions are used to split a document into chunks, sometimes called *pre-tokens*, which are then tokenized separately. Velayuthan & Sarveswaran (2025) recently noted that pre-tokenization can have a greater effect on the resulting tokenization than the choice of tokenization algorithm.

Our cross-domain results in the previous section show that tokenizers produce a common set of tokens across the range of tokenizer training data. We hypothesize that this is due to pre-tokenization: the pre-tokens with highest frequency are prevalent enough that all tokenization algorithms have a strong inclination to find a single token that exactly matches the pre-token. To investigate this, we go back to the aggregated pre-tokens extracted from the corpus with their associated counts. We calculate the proportion of pre-tokens represented as a single token within each tokenizer's vocabulary. Figure 7 displays this proportion for BPE, UnigramLM, and WordPiece tokenizers, for varying vocabulary sizes.

The prevalence of these common pre-tokens within the tokenizers' vocabularies is remarkably high across all algorithms, increasing with vocabulary size. This proportion remains relatively stable with more training data at smaller vocabulary sizes and is essentially overlapping at vocabulary sizes of 128,000 and 256,000. The curves are flat because they represent frequent pre-tokens, which are easily found even with very little training data. Larger vocabularies naturally have more capacity to include a greater number of pre-tokens as single tokens, accounting for the overlapping curves.

These observations support our hypothesis that the plateauing intrinsic metrics (Figure 2) and high weighted Jaccard index (Figure 6) observed with varying training data can at least be partially attributed to the constraints imposed by pre-tokenization. The pre-tokenization step prioritizes the inclusion of commonly occurring pre-tokens as single tokens in the training process. The tokenizers are limited to optimizing the smaller remaining fraction of the tokenized corpus. This limits the tokenizers' ability to fully leverage larger training datasets, as the core vocabulary is largely predetermined by the pre-tokenization process. The additional tokens produced from larger datasets are primarily low-frequency items (i.e., rare words), which have a smaller overall impact on the vocabulary composition and tokeniza-

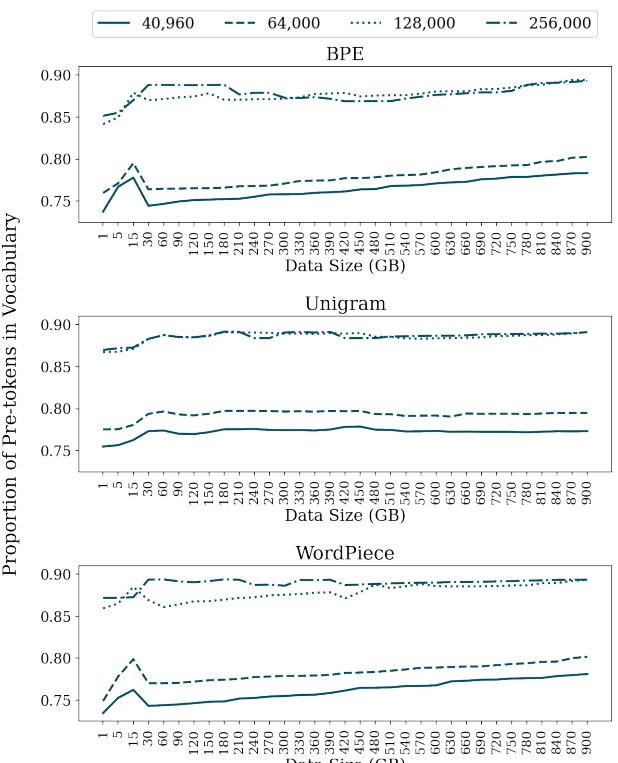

Figure 7: Proportion of pre-tokens represented as single tokens in BPE, UnigramLM and WordPiece vocabularies of varying sizes (40,960, 64,000, 128,000, and 256,000) with increasing training data.

tion quality as measured by our intrinsic metrics. Standard pre-tokenization techniques cannot go beyond word-level segmentation, thereby greatly constraining the vocabulary.

Recent work has begun to explore the influence of pre-tokenization on vocabulary dynamics, offering new perspectives on how pre-token design affects downstream performance and tokenization efficacy (Salehi et al., 2015; Liu et al., 2025; Kumar & Thawani, 2022; Schmidt et al., 2025). These studies highlight the critical role of pre-tokenization in shaping the learned vocabulary and suggest that rethinking or relaxing pre-tokenization constraints could unlock further improvements in tokenizer effectiveness.

## 4. Generalizing To Russian

To better understand whether the impact of tokenizer training data on tokenizer characteristics discussed in this work is robust, we augment our analysis of English data with a focused case study on Russian. Compared to English, Russian is a more morphologically rich language, characterized by extensive inflectional and derivational morphology. This makes it a compelling test case for evaluating how tokeniz-

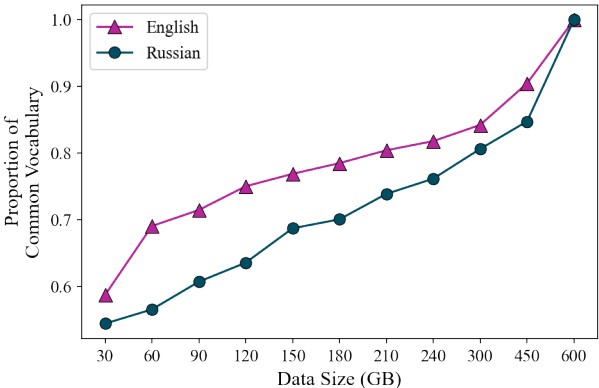

Figure 8: Proportion of common vocabulary for BPE tokenizer (vocabulary size 40,960) trained with cumulatively increasing data, relative to the vocabulary of the corresponding tokenizer trained with 600GB of data.

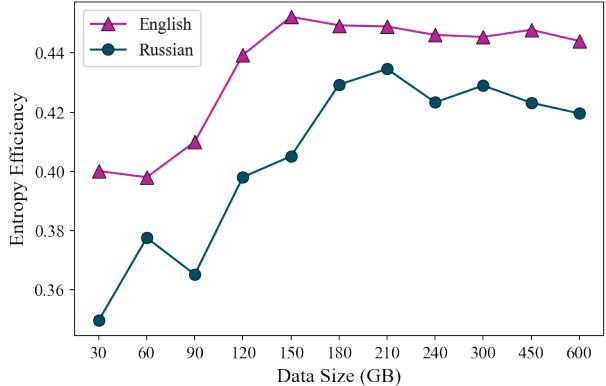

Figure 9: Entropy efficiency on our held-out set for BPE tokenizers (vocabulary size 40,960) with varied training corpus size.

ers handle linguistic complexity under conditions markedly different than those in English.

Our primary goal is to investigate how much training data is required for tokenizers to effectively model such a language. Given that our experiments on English demonstrated relatively stable performance across different tokenizer types and vocabulary sizes, we simplify our exploration by fixing the tokenizer algorithm to BPE and using a vocabulary size of 40,960. We use Russian-language data sourced from the OSCAR dataset (Nguyen et al., 2024), a large multilingual corpus extracted from Common Crawl.[6] Following the cumulative data formation strategy used in earlier experiments, we train tokenizers on increasingly larger subsets of the data. Specifically, we use cumulative data sizes of 30GB, 60GB, 90GB, 120GB, 150GB, 180GB, 210GB, 240GB, 300GB, 450GB, and 600GB. This selection is motivated by observations from our English-language experiments, where both intrinsic metrics (e.g., compression ratio, tokenization efficiency) and Jaccard index on our domain-specific downstream corpus showed saturation beyond typically 150GB and at most 300GB of training data. By extending the upper bound to 600GB for Russian, we aim to determine whether a morphologically richer language exhibits a similar saturation point or requires significantly more data to reach optimal performance. Due to lack of suitable cognitive and morphological evaluation resources, we use entropy efficiency as the primary performance metric in this section.

### 4.1. Vocabulary analysis

As depicted in Figure 8, the proportion of shared vocabulary between tokenizers and a reference tokenizer trained with

---

[6]Version 23.01: `https://oscar-project.github.io/documentation/versions/oscar-2301/`

600GB of data consistently increases with larger Russian training datasets. When juxtaposed with English, we can infer a consistent trend across languages. However, a notable difference is in the rate of increase between the two languages. The rate at which the fraction of common vocabulary increases is comparatively lower for Russian. This could be attributed to the comparatively more complex nature of the language's morphology, where individual types of (mostly) nouns, verbs, and adjectives appear much less frequently than their English equivalents where one form subsumes several, or even dozens, inflected Russian forms.

### 4.2. Entropy analysis

Figure 9 illustrates the entropy efficiency of our tokenizer set, trained on Russian and evaluated on a held-out Russian evaluation set. The figure shows that gains in entropy efficiency become progressively smaller as the training data increases. For English, we observe minimal improvement beyond 150GB of training data. In contrast, for Russian, entropy efficiency continues to improve up to approximately 200GB, after which it stabilizes.

We believe these trends can be attributed to differences in linguistic structure and morphological complexity between the two languages. English, being relatively analytic, has a more limited set of word forms and less inflectional variation, which allows tokenizers to reach near-optimal efficiency with a smaller amount of training data. Once common subword patterns and vocabulary units are sufficiently captured, additional data provides diminishing returns. Russian, on the other hand, is a morphologically rich language with a larger number of inflected forms and derivational variants. This complexity means that a tokenizer needs more data to adequately capture the diverse patterns of morpheme combinations and subword units. As a result, increasing the

training data size continues to yield entropy improvements over a larger range. Beyond this point, the tokenizer has likely encountered the majority of relevant morphological patterns, and additional data no longer provides significant new information, leading to a plateau in entropy efficiency.

Overall, these findings highlight the importance of language-specific considerations when designing or scaling tokenizer training datasets.

## 5. Conclusion

In this work, we systematically investigated the impact of tokenizer training data size on the characteristics of trained tokenizers. Our findings reveal diminishing returns as the tokenizer training data increases beyond 150GB in English, and indications of a limit closer to 200GB in Russian, suggesting some dependence on the morphological properties of the language being tokenized. Our analysis indicates that tokenization algorithms incorporating pre-tokenization may be fundamentally limited in their ability to fully leverage extremely large datasets. Therefore, rather than focusing solely on *more data*, we advocate for a shift towards developing and employing better vocabulary training methods that are less susceptible to the limitations of pre-tokenization. Methods that tokenize beyond pre-token boundaries (Liu et al., 2025; Schmidt et al., 2025), and ones that incorporate contextual signals (Yehezkel & Pinter, 2023), offer promising directions for future research. Their main drawback—nonlinear dependence on data size—is now believed to be less problematic than previously assumed.

## Limitations

In this study, we use intrinsic tokenizer metrics as a primary means of evaluation, while enabling efficient analysis, may not fully capture the complex interplay between tokenizer characteristics and downstream LLM performance. While these metrics provide valuable insights into tokenizer quality, they serve as a proxy for true downstream effectiveness. Future research should investigate the correlation between intrinsic metrics and downstream task performance across a wider range of language models and tasks to establish a more comprehensive evaluation framework.

## Ethics Statement

We train our tokenizers on the commonly used public datasets The Pile (Gao et al., 2020), RedPajama (Weber et al., 2024) and OSCAR (Nguyen et al., 2024), which have not undergone a formal ethics review. While our evaluation set was manually anonymized and checked for abusive language, it may still contain personal opinions that reflect cultural, political, or demographic biases.

## Acknowledgments

We thank Seth Ebner for many notes and discussions. This research was supported in part by the Israel Science Foundation (grant No. 1166/23).

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

## A. Pre-tokenization regular expression

We repeat the pre-tokenization regular expression used in our implementation, and explain each of its branches:

```
r"(?i:[sdmt]|ll|ve|re)|[^\r\n\p{L}\p{N}]?+\
  p{L}+|\p{N}{1,3}|␣?[^\s\p{L}\p{N}]++[\r
  \n]|\s*[\r\n]|\s+(?!\S)|\s+"
```

Listing 2: GPT-4 pre-tokenizer regular expression (repeated).

- `(?i:[sdmt]|ll|ve|re)`

  Matches English language contractions or possessives

- `[^\r\n\p{L}\p{N}]?+\p{L}+`

  Matches one or more Unicode letters, optionally preceded by a punctuation or symbol character

- `\p{N}{1,3}`

  Matches one to three Unicode digits (some languages such as Thai and Tamil use different Unicode symbols for digits)

- ` ?[^\s\p{L}\p{N}]++[\r\n]*`

  Matches one or more punctuation or symbol characters, optionally preceded by a space, and succeeded by zero or more carriage returns or linefeeds

- `\s*[\r\n]`

  Matches a single carriage return or linefeed, optionally preceded by zero or more characters of whitespace

- `\s+(?!\S)`

  Matches one or more characters of whitespace not immediately followed by non-whitespace, thus matching trailing whitespace of a line, up to the line break

- `\s+`

  Finally, this matches one or more characters of whitespace

This regular expression requires the more flexible `regex` package in Python rather than the default `re` package, in order to support Unicode character groups and the possessive quantifiers `?+` and `++`.

Note that this expression has the necessary property to match all characters in any valid UTF-8. This is because `[^\s\p{L}\p{N}]` in the fourth branch will match any character that is not a letter, number, or whitespace, which are all handled by other parts of the regex. Dagan et al. (2024) had a similar explanation of this same regex, plus other possible regular expression choices.[7]

---

[7]See https://tokencontributions.substack.com/p/pre-tokenization-on-punctuation-in for other discussions on peculiarities of this particular regular expression.

## B. Analysis of Common Vocabulary

Figures 10, 11, and 12 present heatmaps visualizing the proportion of shared vocabulary among all trained tokenizers for BPE, UnigramLM, and WordPiece, respectively. Within each figure, subplots correspond to different vocabulary sizes: 40,960, 64,000, 128,000, and 256,000.

A key observation for both BPE (Figure 10) and UnigramLM (Figure 11) is the decreasing proportion of shared vocabulary between consecutively trained tokenizers as the vocabulary size increases. For instance, with BPE and a 40,960 vocabulary, the vocabulary overlap between tokenizers trained on 30GB and 900GB of data is 0.57. This overlap decreases to 0.49, 0.41, and 0.23 for vocabulary sizes of 64,000, 128,000, and 256,000, respectively. UnigramLM tokenizers exhibit a similar trend, with overlap decreasing from 0.58 to 0.51, 0.34, and 0.2 across the same vocabulary sizes.

This trend suggests that as vocabulary size increases for a fixed training data size (e.g., 30GB), the added tokens become progressively less frequent and more specialized. These less frequent or more niche tokens are also more susceptible to variation between tokenizers trained on slightly different subsets of data drawn from the same overall distribution. Essentially, with limited training data, larger vocabularies are populated with progressively less consequential tokens, leading to lower agreement between trained tokenizers. However, WordPiece (Figure 12) does not exhibit any trend with a change in vocabulary size.

## C. Analysis on Evaluation Data

### C.1. Data Source

Our tokenizer training data spans from 2021 to early 2023. To prevent data leakage into our evaluation set, we sourced documents from various domains specifically dated 2024. We manually de-anonymized and removed any offensive language from this dataset. The following describes the composition of our evaluation data for each domain:

- *Biology*: We utilized research papers and clinical trial data from the National Library of Medicine.[8]

- *Code*: Code was generated using GPT-4 for randomly selected programming problems drawn from IIT[9] homework assignments and LeetCode.[10] The code corpus contains a mixture of popular programming languages including Python, Java, JavaScript, Ruby, and Go, further enhancing diversity.

---

[8]https://www.ncbi.nlm.nih.gov/

[9]https://www.cse.iitk.ac.in/users/nitin/courses/CS681-2019-20-II/problemsets.html

[10]https://leetcode.com/

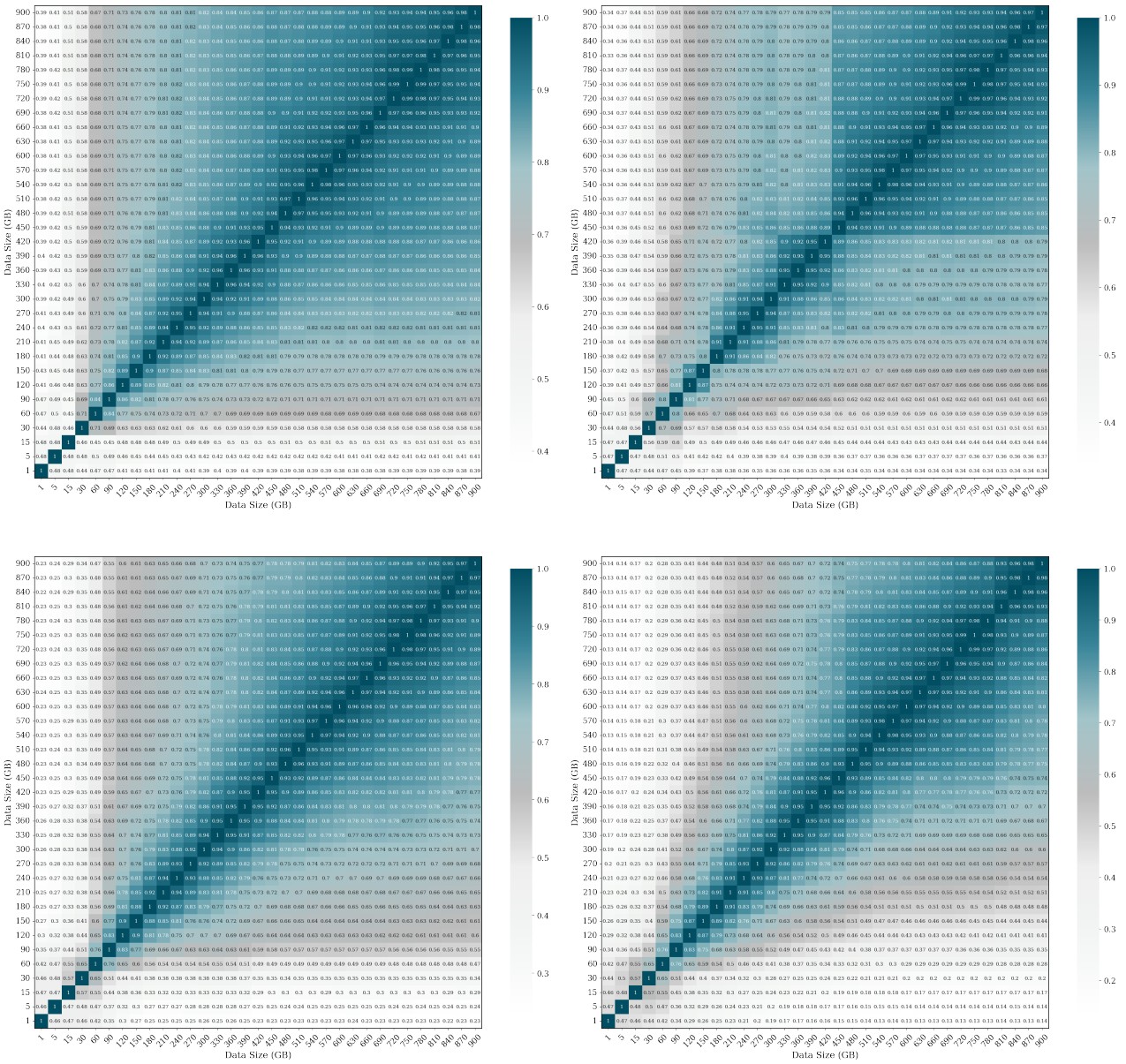

Figure 10: Heatmaps showing the proportion of common vocabulary across all **BPE** tokenizers as a function of cumulatively increasing training data. Each heatmap, in the grid from the top left, represents a different vocabulary size (40,960, 64,000, 128,000, 256,000).

- *Finance*: This dataset comprises SEC filings[11] from various companies filed in 2024, downloaded and extracted from PDF to text format.

- *History*: We used papers from Oxford Academic Historical Research. [12]

- *Legal*: This dataset consists of 2024 - Opinions of the Court released by the Supreme Court of the United States.[13]

- *Math*: Mathematics papers from arXiv,[14] all released in 2024, were used.

- *General Conversation*: We employed general conversation data from the same population as our tokenizer training data, ensuring no overlap between training and evaluation sets.

---

[11] https://www.sec.gov/

[12] https://academic.oup.com/histres/

[13] https://www.supremecourt.gov/opinions/slipopinion/24

[14] https://arxiv.org/archive/math

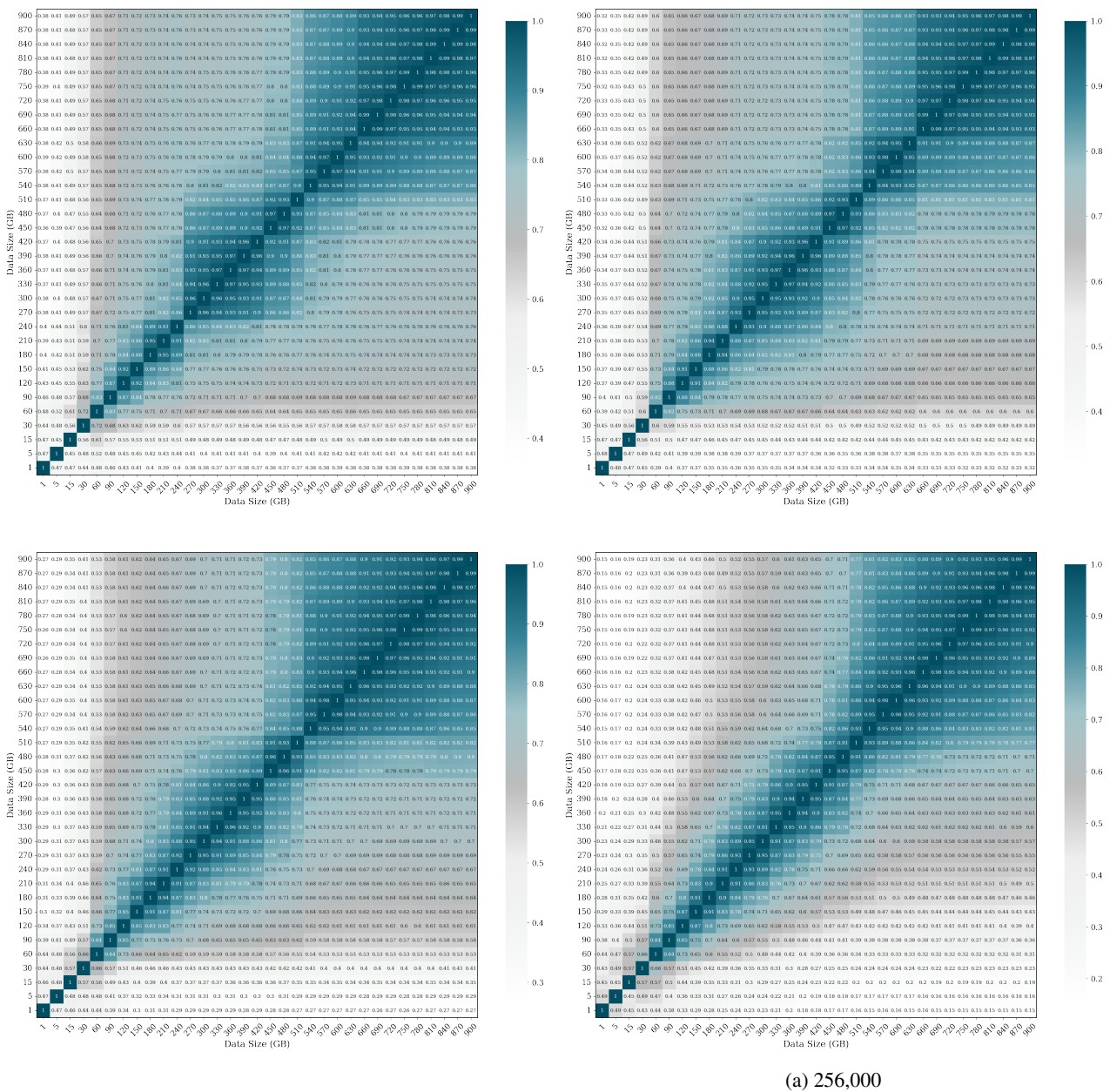

(a) 256,000

Figure 11: Heatmaps showing the proportion of common vocabulary across all **UnigramLM** tokenizers as a function of cumulatively increasing training data. Each heatmap, in the grid from the top left, represents a different vocabulary size (40,960, 64,000, 128,000, 256,000).

## C.2. Analysis

As mentioned in Section 2.3, we assess the impact of scaling training data for tokenizer training on the evaluation set by computing the Jaccard Index and weighted version to the Jaccard Index between the actual tokens used in the evaluation text using each trained tokenizer and the same text tokenized with the 900GB-trained reference tokenizer. Within the evaluated vocabulary size of 40,960 (see Figure 13), slightly different trends emerge across different tasks. However, both BPE and UnigramLM tokenizers exhibit a near-immediate plateau in performance, around 120GB to 180GB. This suggests that, at this vocabulary size, frequent tokens contribute substantially to the overall token overlap and that these tokenizers quickly converge to a stable segmentation of these frequent terms. This early plateau indicates that further increases in training data provide diminishing returns for these tokenization algorithms.

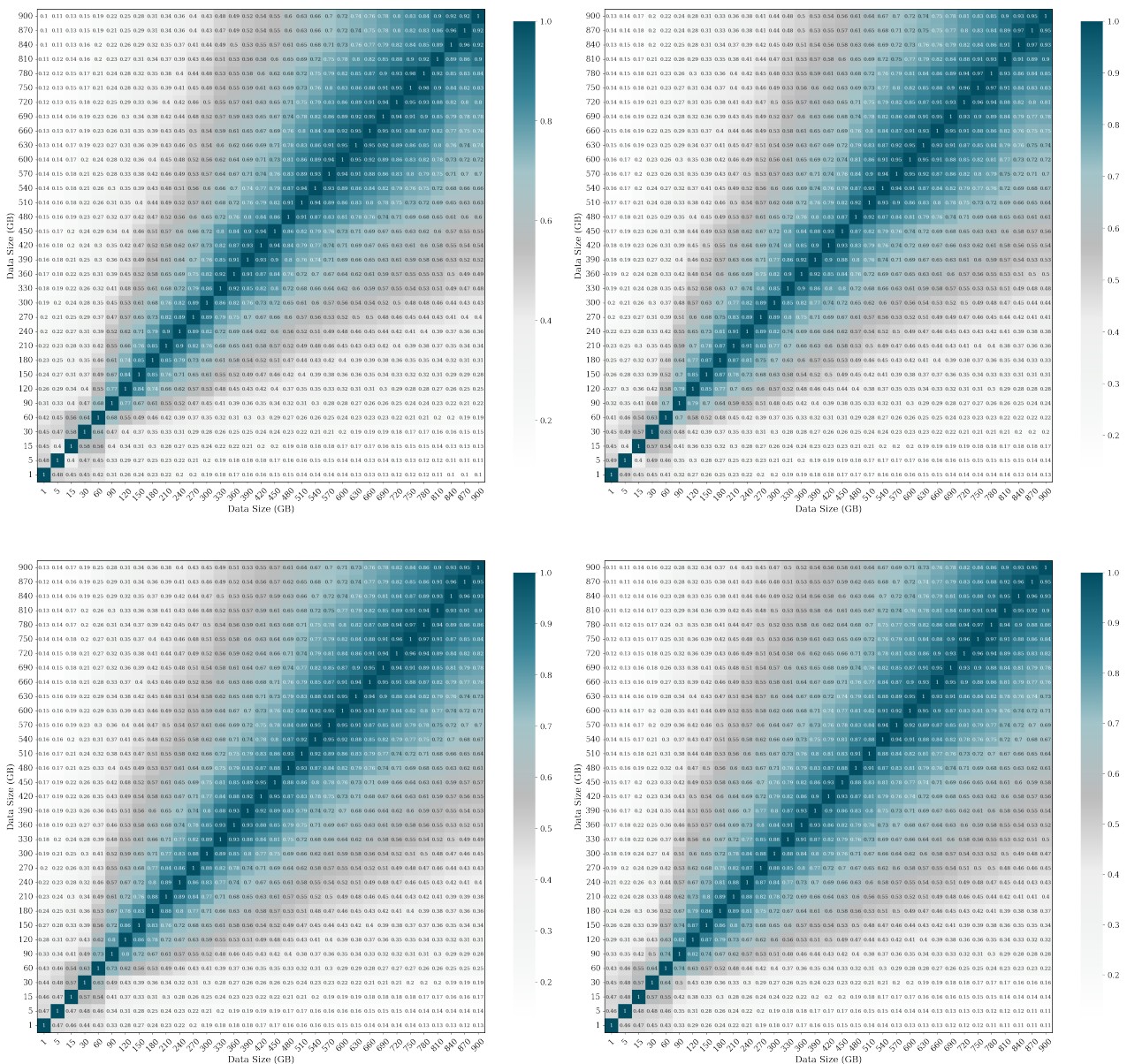

Figure 12: Heatmaps showing the proportion of common vocabulary across all **WordPiece** tokenizers as a function of cumulatively increasing training data. Each heatmap, in the grid from the top left, represents a different vocabulary size (40,960, 64,000, 128,000, 256,000).

Although WordPiece demonstrates a similar trend of a much lower Jaccard Index compared to its weighted counterpart, it exhibits the widest gap between the two Jaccard Index values. A large difference between the Jaccard Index and the Weighted Jaccard Index indicates that the overlap between vocabulary usage is primarily due to less frequent tokens, while the more frequent tokens are not consistently shared. This could mean that the change in vocabulary might have higher implications for the tokenization of text, relative to BPE and Unigram LM. These patterns remain consistent

across WordPiece for the remaining vocabulary sizes, as seen in Figures 13 to 16.

### C.3. Computing Domain-Specific and General English Terms

To identify and quantify domain-specific and general English terms within a corpus, we use a comparative frequency-based approach grounded in corpus linguistics. The core idea is to contrast the term distributions between a domain-specific corpus ($D$) and a general English corpus ($G$) to

determine which terms are characteristic of the domain.

We curate two text corpora: a domain corpus ($D$), which is a compilation of all our downstream domain-specific evaluation data, and a general corpus ($G$),[15] representing standard English usage. Both corpora are matched in size to ensure a fair comparison.

For each corpus, we compute the relative term frequency of each word $w$:

$$\text{TF}_D(w) = \frac{\text{count}(w \in D)}{\sum_{w'} \text{count}(w' \in D)},$$

$$\text{TF}_G(w) = \frac{\text{count}(w \in G)}{\sum_{w'} \text{count}(w' \in G)}$$

To measure domain relevance, we compute a score for each term based on its relative frequency in both corpora:

$$\text{Score}(w) = \frac{P(w|D)}{P(w|G) + \epsilon}$$

where $P(w|D)$ and $P(w|G)$ are the relative frequencies of $w$ in the domain and general corpora, respectively, and $\epsilon$ is a small smoothing constant to avoid division by zero.

A higher score indicates stronger domain specificity. Terms with extremely low counts (e.g., fewer than five occurrences across both corpora) are filtered out to reduce noise. Based on this score, we classify terms as follows:

- **Domain-Specific Terms:** $\text{Score}(w) \geq \theta$

- **General English Terms:** $\text{Score}(w) \leq \frac{1}{\theta}$

- **Neutral Terms:** $\frac{1}{\theta} < \text{Score}(w) < \theta$

For the threshold, we use $\theta = 10$. This methodology enables us to quantify the number of domain-specific and general English terms present in a given vocabulary.

We group general English and neutral terms together. After classification, we manually inspect and alter, whenever necessary, the resulting word lists to ensure that there is no contamination between domain-specific and general terms.

---

[15]https://huggingface.co/datasets/wikimedia/
wikipedia

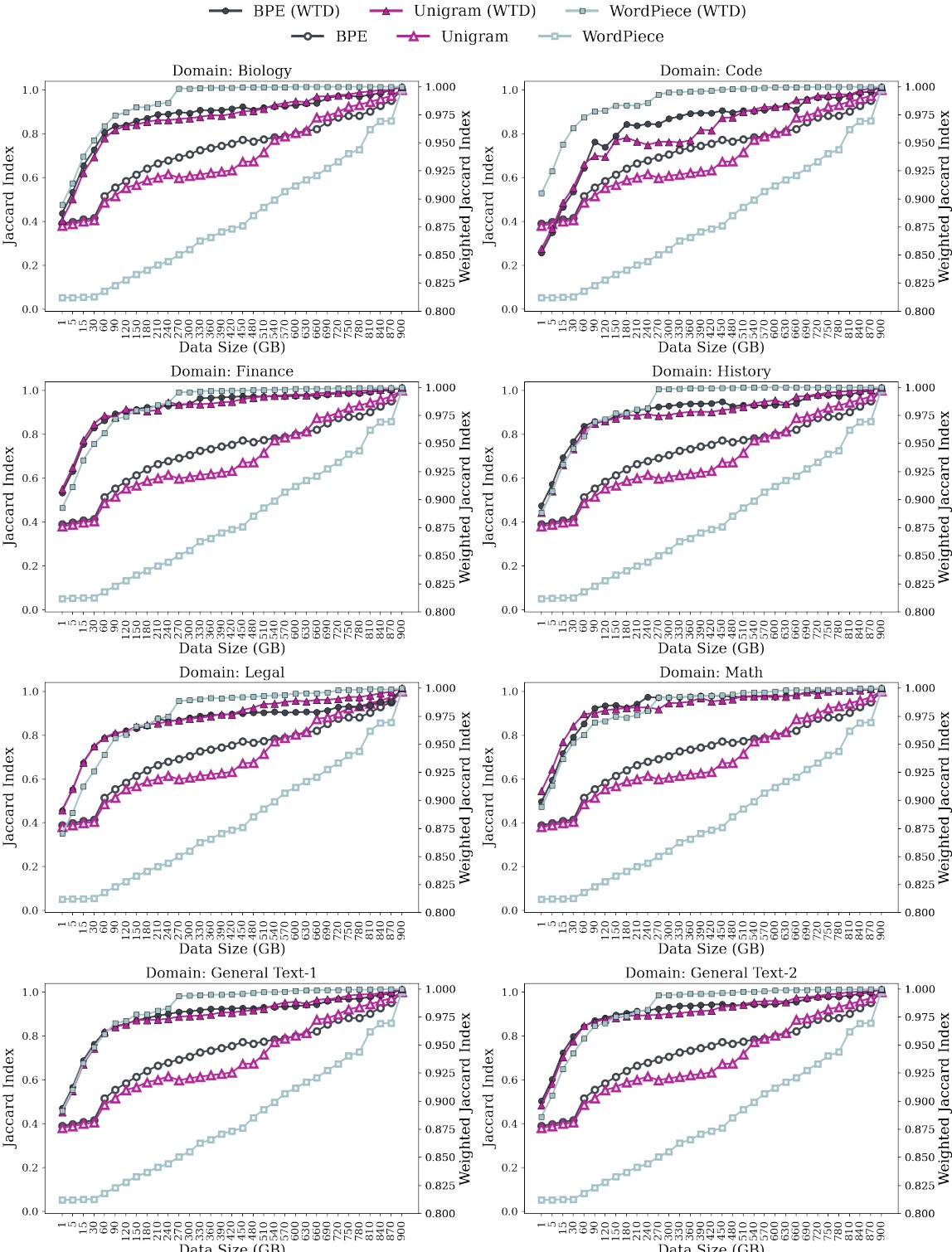

Figure 13: Jaccard Index (open markers) and Weighted Jaccard Index (WTD; filled markers) for BPE, UnigramLM, and WordPiece tokenizers (vocab size **40,960**) across varying data sizes, for different domains.

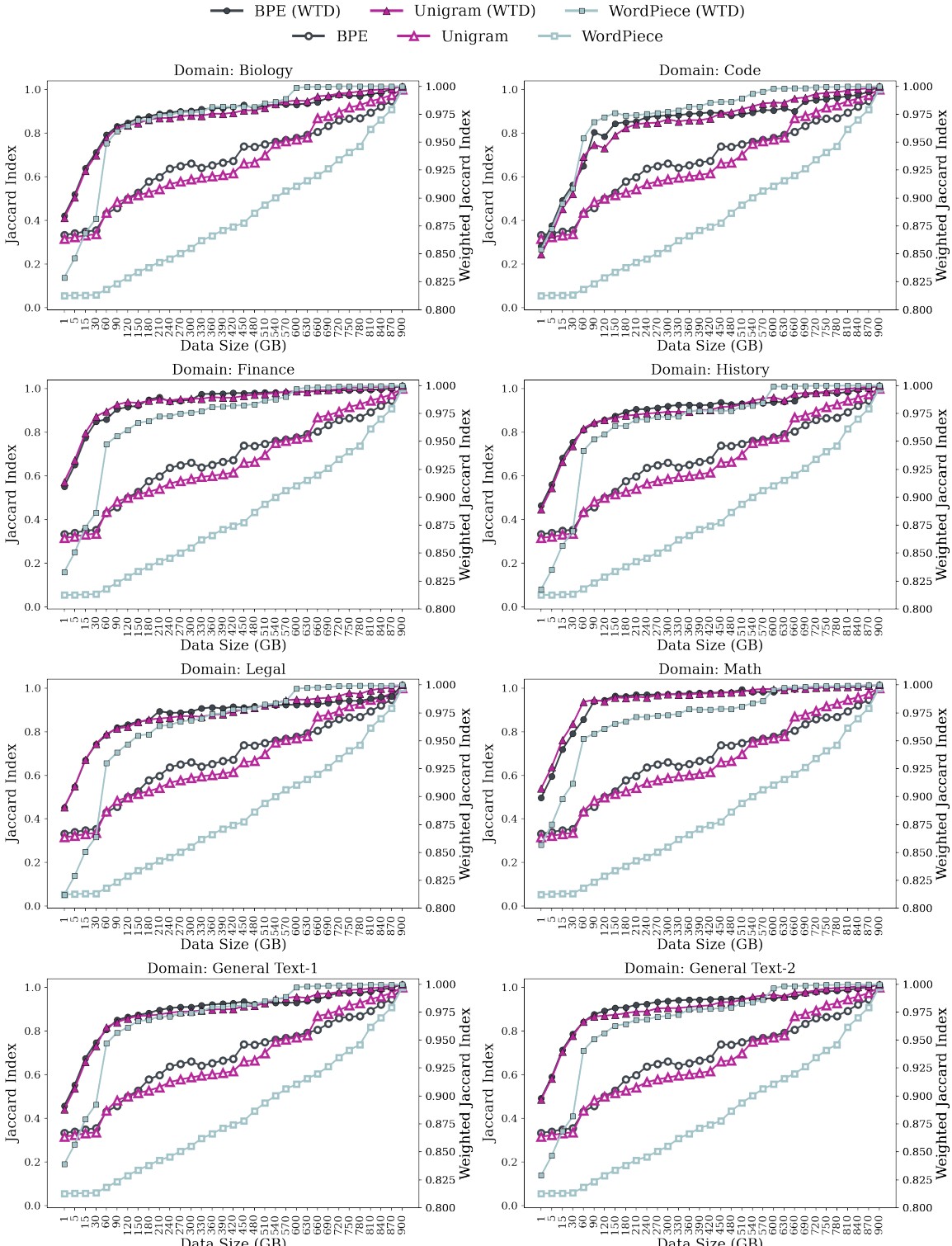

Figure 14: Jaccard Index (open markers) and Weighted Jaccard Index (WTD; filled markers) for BPE, UnigramLM, and WordPiece tokenizers (vocab size **64,000**) across varying data sizes, for different domains.

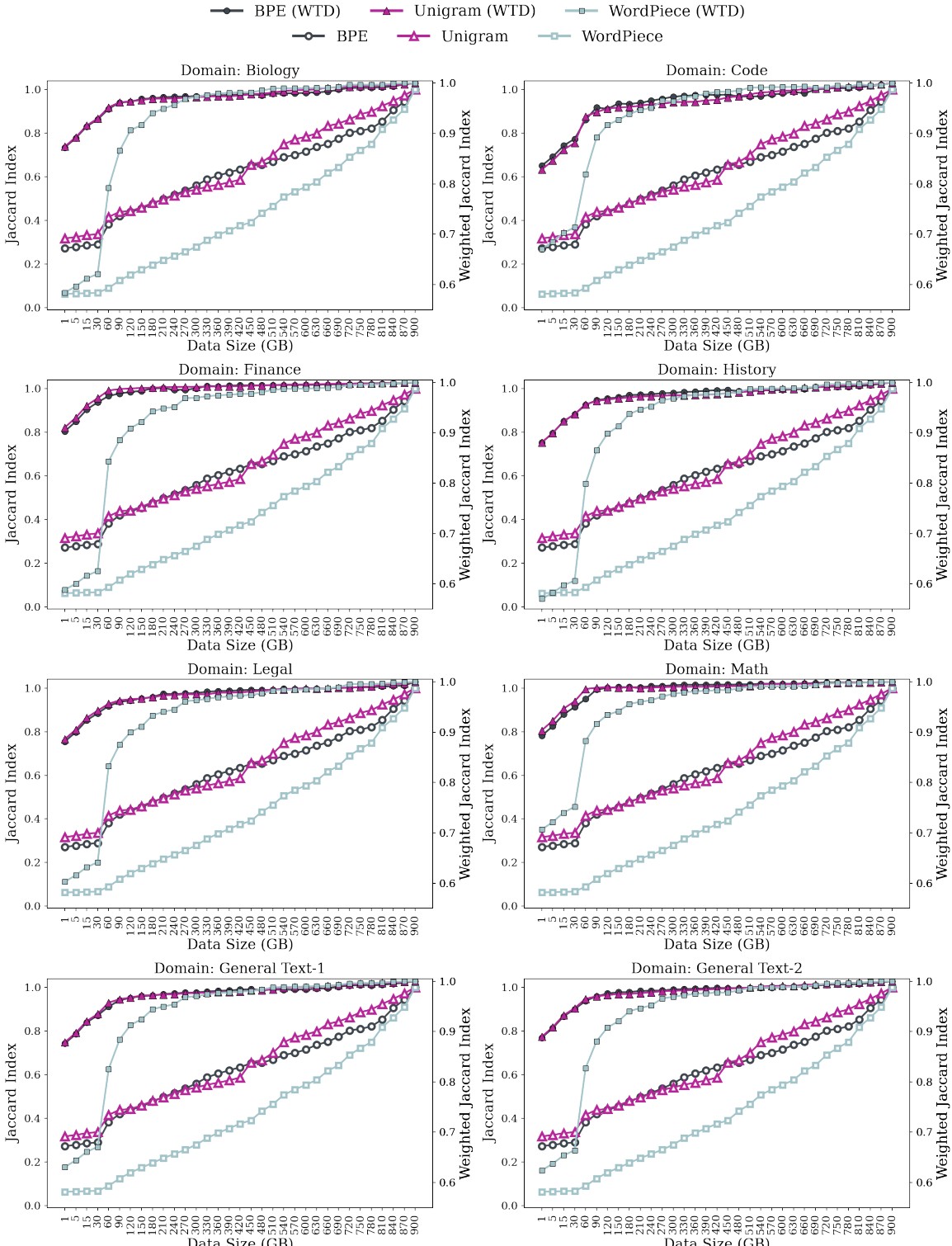

Figure 15: Jaccard Index (open markers) and Weighted Jaccard Index (WTD; filled markers) for BPE, UnigramLM, and WordPiece tokenizers (vocab size **128,000**) across varying data sizes, for different domains.

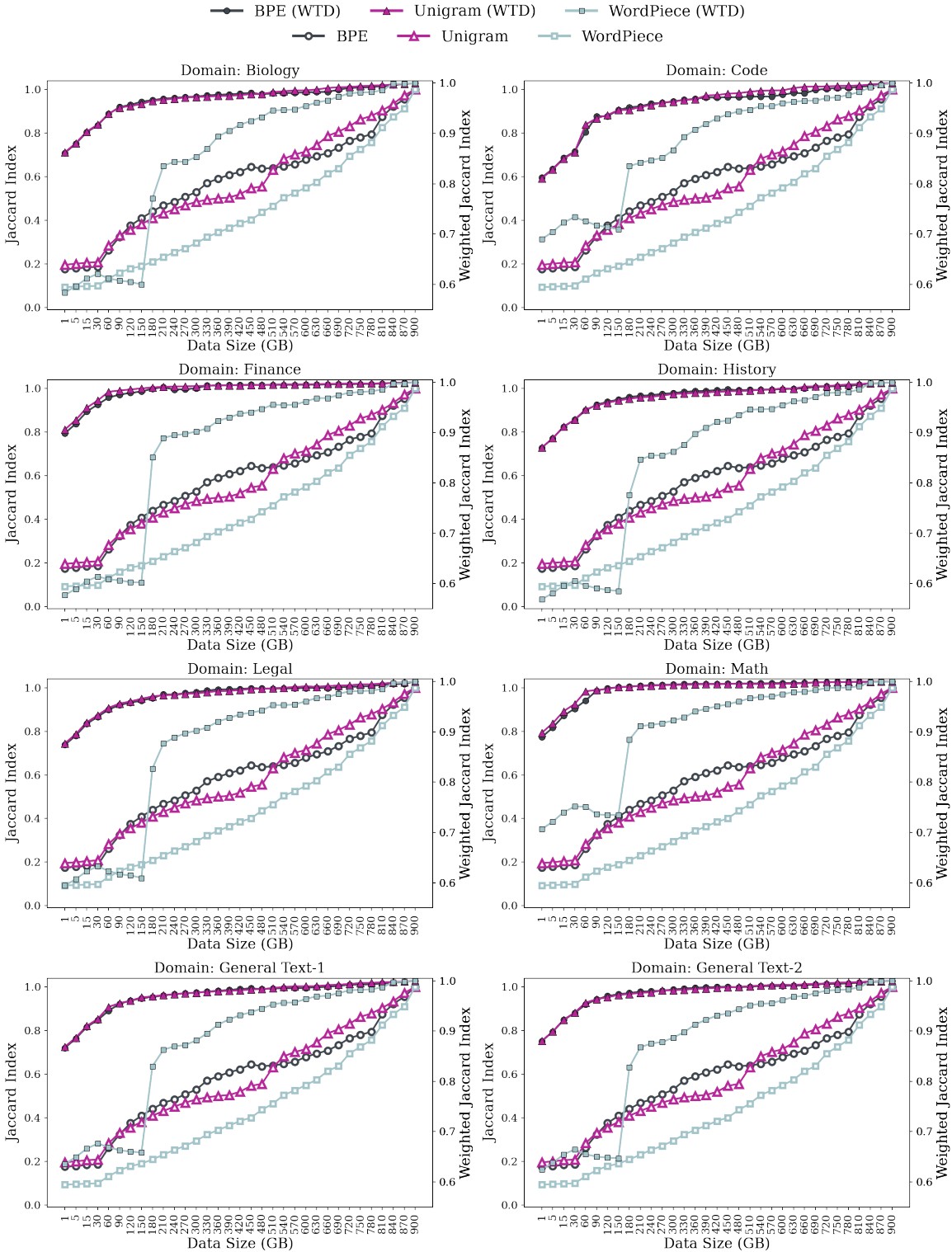

Figure 16: Jaccard Index (open markers) and Weighted Jaccard Index (WTD; filled markers) for BPE, UnigramLM, and WordPiece tokenizers (vocab size **256,000**) across varying data sizes, for different domains.

