# OpenReview forum: "How Much is Enough? The Diminishing Returns of Tokenization Training Data"
_ICML.cc/2025/Workshop/TokShop — TokShop_

### Official Review · Reviewer_ySmv · 2025-06-05
**Thorough study of tokenizer training data saturation**

**Rating:** 7
**Confidence:** 4

**Review:**

This paper investigates the impact of tokenizer training data size on the quality and characteristics of the resulting tokenizers.
The paper posits that this saturation is largely due to the constraints imposed by pre-tokenization steps, though saturation happens at fairly large data sizes compared to what is commonly used.

Strengths:
- Addresses a practical and relevant question regarding optimal tokenizer training data requirements.
- Clear demonstration of diminishing returns for English and Russian, with a plausible explanation.
- The investigation is thorough, at least for English, presenting extensive results across multiple algorithms and vocabulary sizes.

Weaknesses:
- The evaluation is limited to two space-separated languages (English, Russian). Given the paper's emphasis on pre-tokenization's role, an analysis of a language like Chinese, which lacks explicit word delimiters, would have significantly strengthened this argument.
- The paper is quite long for such a narrow topic, and could be clearer in highlighting and presenting the key messages.

---

### Official Review · Reviewer_fCLC · 2025-06-06
**Review of Submission 35**

**Rating:** 9
**Confidence:** 5

**Review:**

This paper tackles an extremely important issue, as there have been very few systematic studies about how to best train tokenizers. The paper is clear and the experiments are carefully manipulated.

In Section 2.2, the explanations of the metrics are overly brief. I suggest adding more detail to help the reader build up an intuition about what these metrics tell us about a tokenizer. In this section, engage more with previous work.

I also recommend changing the name of Section 4 to `Generalizing to Russian’. The paper makes too strong of a claim about generalizing to other languages, especially as the authors did not test multiple vocabulary sizes for Russian. I think the authors should also minimize the extent to which they characterize Russian as a morphologically complex language. While more complex than English, it does not have as complex a morphological system as languages like Turkish or Hungarian. It is not really appropriate to draw such strong conclusions about morphology, given that English and Russian aren’t that different and there are other differences between the languages, such as writing system.

Questions:
* Do you think that it’s possible that training on too much text could lead to over-training?
* Are you going to release the tokenizers?
* Do you have any idea about what is happening with the peaks in Figures 1 and 7?
* Why do you think the smallest vocab have the best scores on the metrics from Uzan et al. (2024)? You don’t compare them to compression in Fig. 1, but I presume then that there is not a positive correlation between compression and those metrics? Could it be that the morphological alignment score is driven by oversegmentation?

Suggestions
* For future work, use MorphScore to evaluate morphological alignment. MorphScore v2 includes Russian. https://github.com/catherinearnett/morphscore/tree/v2
* The authors should train tokenizers with different vocab sizes for Russian or other languages and test whether the patterns are the same as English with respect to vocab size.

---

### Official Review · Reviewer_xjnJ · 2025-06-08
**Tokenization at Scale: When More Data Stops Helping**

**Rating:** 6
**Confidence:** 4

**Review:**

## Paper Summary
The paper studies the impact of the size of tokenizer training data, a poorly studied NLP parameter, on tokenization performance. The authors conduct an extensive empirical study with BPE, UnigramLM, and WordPiece tokenizers trained on English data up to 900GB, and Russian data up to 600GB. They test vocabulary overlap, intrinsic evaluations, and cross-domain performance. The main finding is that adding training data beyond ~150GB (English) and ~200GB (Russian) provides diminishing returns in tokenization quality. The paper also looks into how far the limitations of pre-tokenization hold back improvement, even with additional data, and suggests that crossing these thresholds is prerequisite to actual gains.

## Strengths
1. Novelty: Focus on tokenizer training data size is novel and addresses a less explored but instrumentally important issue.
2. Comprehensiveness: Experimental design is exhaustive, including different types of tokenizers, vocabulary sizes, and languages.
3. Readability of Results: Results are readable, with good visualizations and adequately motivated analysis.
4. Actionable Insights: The research delivers tangible recommendations for resource allocation in tokenizer and LLM development, and proposes strong avenues for future work.

## Weaknesses
- The paper is grounded entirely in intrinsic measures. While as rationalized as this would have needed to be in terms of cost of downstream tests, including even a one small extrinsic test (e.g., task accuracy or perplexity) would have made the claims stronger.
- The experiments use a single pre-tokenizer (a GPT-4-inspired regex). As pre-tokenization is identified as a bottleneck, some comparison or ablation here would be helpful.
- While the Russian experiments are a nice contribution, other highly morphologically dense or non-Latin languages can serve to further validate the generalization assertions.
- There's not much discussion of multilingual tokenizers or multilingual models trained over many different languages, which is ever more important in real-world practice.

## Recommendation: Weak Accept
**Justification**: This is a good empirical contribution and resolves a practical question that has been surprisingly under-researched. While some limitations — most notably the absence of downstream tests — there is no methodological flaw, results are well-supported, and conclusions are useful for researchers as well as practitioners. The paper raises some interesting questions about the boundaries of current tokenization pipelines and leaves some space for future innovation. Well-suited for a workshop presentation.

## Reviewer Confidence: 4 / 5
I am fairly familiar with the topic and feel confident in assessing the strengths and limitations of this work.

---

### Decision · Program_Chairs · 2025-06-10

Accept